# Bi-allelic pathogenic variants in *NDUFC2* cause early-onset Leigh syndrome and stalled biogenesis of complex I

Ahmad Alahmad[1,2,3], Alessia Nasca[4] , Juliana Heidler[5], Kyle Thompson[1,2], Monika Oláhová[1,6], Andrea Legati[4], Eleonora Lamantea[4], Jana Meisterknecht[5], Manuela Spagnolo[4], Langping He[1,7], Seham Alameer[8,9,10], Fahad Hakami[11], Abeer Almehdar[12], Anna Ardissone[13], Charlotte L Alston[1,2,7], Robert McFarland[1,2,7], Ilka Wittig[5,14,†], Daniele Ghezzi[4,15,†] & Robert W Taylor[1,2,7,†,*]

## Abstract

**Leigh syndrome is a progressive neurodegenerative disorder, most commonly observed in paediatric mitochondrial disease, and is often associated with pathogenic variants in complex I structural subunits or assembly factors resulting in isolated respiratory chain complex I deficiency. Clinical heterogeneity has been reported, but key diagnostic findings are developmental regression, elevated lactate and characteristic neuroimaging abnormalities. Here, we describe three affected children from two unrelated families who presented with Leigh syndrome due to homozygous variants (c.346_*7del and c.173A>T p.His58Leu) in *NDUFC2*, encoding a complex I subunit. Biochemical and functional investigation of subjects' fibroblasts confirmed a severe defect in complex I activity, subunit expression and assembly. Lentiviral transduction of subjects' fibroblasts with wild-type *NDUFC2* cDNA increased complex I assembly supporting the association of the identified *NDUFC2* variants with mitochondrial pathology. Complexome profiling confirmed a loss of NDUFC2 and defective complex I assembly, revealing aberrant assembly intermediates suggestive of stalled biogenesis of the complex I holoenzyme and indicating a crucial role for NDUFC2 in the assembly of the membrane arm of complex I, particularly the ND2 module.**

**Keywords** complex I; Leigh syndrome; mitochondrial disease; NDUFC2; OXPHOS

**Subject Categories** Genetics, Gene Therapy & Genetic Disease; Organelles

See also: **K Reinson & K Õunap** (November 2020)

## Introduction

Mitochondrial disorders are a group of complex genetic disorders characterised by mitochondrial dysfunction, resulting in clinically heterogeneous presentations mostly affecting tissues with high energy requirements (Gorman *et al*, 2016). Mitochondrial proteins are encoded by either the nuclear genome (involving > 1,000 genes) or the mitochondrial genome (13 protein-coding genes), with pathogenic variants in more than 330 genes having been associated with mitochondrial disorders to date (Thompson *et al*, 2020). Due to this genetic heterogeneity, mitochondrial disease can follow any mode of inheritance including maternal, autosomal (dominant or recessive) or X-linked inheritance or occur *de novo*. Groups of frequently reported clinical manifestations associated with mitochondrial

1 Wellcome Centre for Mitochondrial Research, Newcastle University, Newcastle upon Tyne, UK
2 Translational and Clinical Research Institute, Faculty of Medical Sciences, Newcastle University, Newcastle upon Tyne, UK
3 Kuwait Medical Genetics Centre, Al-Sabah Medical Area, Kuwait
4 Unit of Medical Genetics and Neurogenetics, Fondazione IRCCS Istituto Neurologico Carlo Besta, Milan, Italy
5 SFB815 Core Unit, Functional Proteomics, Medical School, Goethe-Universität, Frankfurt am Main, Germany
6 Faculty of Medical Sciences, Biosciences Institute, Newcastle University, Newcastle upon Tyne, UK
7 NHS Highly Specialised Service for Rare Mitochondrial Disorders, Royal Victoria Infirmary, Newcastle upon Tyne Hospitals NHS Foundation Trust, Newcastle upon Tyne, UK
8 Pediatric Department, Ministry of National Guard Health Affairs, Jeddah, Saudi Arabia
9 King Abdullah International Medical Research Center, Jeddah, Saudi Arabia
10 King Saud bin Abdulaziz University for Health Sciences, Jeddah, Saudi Arabia
11 Section of Molecular Medicine, King Abdulaziz Medical City-WR, King Saud bin Abdulaziz University for Health Sciences, Jeddah, Saudi Arabia
12 Department of Medical Imaging, King Saud bin Abdulaziz University for Health Sciences, King Abdulaziz Medical City-WR, National Guard Health Affairs, Jeddah, Saudi Arabia
13 Child Neurology, Fondazione IRCCS Istituto Neurologico Carlo Besta, Milan, Italy
14 German Center for Cardiovascular Research (DZHK), Partner site RheinMain, Frankfurt, Germany
15 Department of Pathophysiology and Transplantation, University of Milan, Milan, Italy
 *Corresponding author. Tel: +44 0191 208 3685; E-mail: robert.taylor@ncl.ac.uk
 †These authors contributed equally to this work

disease have been classified as syndromes, one of which is Leigh syndrome.

Leigh syndrome is the most commonly reported neurodegenerative disorder in paediatric mitochondrial disease (Rahman et al, 2017). Defects involving more than 90 genes are associated with Leigh syndrome, implicating both the nuclear and mitochondrial genomes (Lake et al, 2019). Clinical heterogeneity has been reported in Leigh syndrome patients, but key findings include delayed development, loss of acquired milestones, hypotonia, elevated cerebrospinal fluid (CSF) lactate and multisystem involvement. Radiological imaging typically reveals bilateral and symmetrical involvement of the basal ganglia and/or brainstem (Rahman et al, 1996).

More than a third of reported Leigh syndrome cases presented with complex I (NADH:ubiquinone oxidoreductase) deficiency (OMIM: 252010), a common biochemical phenotype in paediatric mitochondrial disease (Lake et al, 2016, 2017; Rahman et al, 2017). As the largest of the mitochondrial respiratory chain complexes, complex I comprises 44 different subunits with at least 14 assembly factors required for its biosynthesis (Stroud et al, 2016; Zhu et al, 2016; Guerrero-Castillo et al, 2017; Formosa et al, 2020). Studying the effect of systematically knocking out each accessory complex I subunit has revealed defective or stalled complex I assembly, establishing the modular nature of complex I assembly (Stroud et al, 2016). Additionally, complexome profiling of patient cell lines with genetically characterised mitochondrial complex I deficiency has been exploited to better understand the molecular mechanisms underlying assembly and stability in vivo (Alston et al, 2016, 2018, 2020).

To date, pathogenic variants in 42 genes encoding either complex I subunits or assembly factors have been associated with isolated complex I deficiency, of which 27 have been identified in Leigh syndrome patients (Rahman et al, 2017; Rouzier et al, 2019; Alston et al, 2020; Thompson et al, 2020). Due to the clinical and genetic heterogeneity within patients presenting with Leigh syndrome—and indeed mitochondrial disorders as a whole—genetic diagnoses are often difficult to achieve. Next-generation sequencing, in the form of unbiased whole exome/genome sequencing (WES/WGS), and also targeted gene panels have been fruitful in improving the diagnostic rate in mitochondrial disorders and led to the discovery of novel candidate genes underlying mitochondrial disease (Legati et al, 2016; Abicht et al, 2018; Theunissen et al, 2018). The application of WES to cohorts of mitochondrial disease patients has reported causative genetic variants in approximately 60% of cases (Taylor et al, 2014; Wortmann et al, 2015; Pronicka et al, 2016).

This report highlights the clinical findings of three paediatric subjects from two unrelated consanguineous families who presented with Leigh syndrome due to rare, homozygous variants in NDUFC2, which encodes a structural complex I subunit. Our study demonstrates the importance of data sharing in rare diseases and the biochemical and molecular experimental validation of novel variants identified by next-generation sequencing; at present, the study of human patient cell lines is fundamental for validating pathogenicity but, in addition, can be extremely useful for improving our understanding of mitochondrial biology, such as complex I assembly pathways.

# Results

## Case reports

Subject 1, a female infant, is the second child born to healthy consanguineous first cousin Saudi parents (Family 1; Fig 1A). She was born by normal vaginal delivery at 37-week gestation (birthweight 2.0 kg, 2nd–9th centile) with no antenatal issues. Developmental delay was noted at 2 years when she was unable to walk. At 6 years, her height and weight remained below the 5th centile and clinical examination revealed facial dysmorphic features, spasticity and brisk deep tendon reflexes. She was unable to stand without support and unable to speak. A hearing assessment was unremarkable, but an ophthalmological examination revealed bilateral optic disc pallor. She did not develop seizures and had no cardiac, renal or hepatic abnormalities. Serum lactate ranged from 3.6 to 7.6 mmol/l (normal range 0.7–2.2 mmol/l), whilst ammonia, blood glucose and thyroid function were unremarkable. Plasma amino acid analysis detected elevated alanine (724 µmol/l, normal range 143–439 µmol/l) and elevated proline (366 µmol/l, normal range 52–298 µmol/l). Urine organic acid analysis showed increased fumaric acid excretion (151 mM/M creatinine, normal range < 20 mM/M). Leucocyte enzyme assay for arylsulfatase A, arylsulfatase B and galactocerebrosidase was unremarkable, as was abdominal ultrasound and array comparative genomic hybridisation (aCGH) testing. Brain MRI at 21 months showed bilateral areas of abnormal high signal intensity at the corticospinal tract level of the corona radiata with loss of white matter volume at these areas. This was associated with irregularity in the outline of the lateral ventricles suggestive of periventricular leukomalacia. Bilateral symmetrical abnormal high T2 signals of the medial aspect of the thalami, substantia nigra and posterior tract of the medulla oblongata were also identified (Fig EV1A and B).

Subject 2, was the younger brother of Subject 1; foetal echocardiogram showed mild cardiomegaly, dilated superior vena cava and a small ventricular septal defect (VSD). Antenatal ultrasound showed dilated cisterna magna and ventriculomegaly. He was born by normal vaginal delivery at 36-week gestation (birthweight 2.020 kg, 2nd centile). Postnatal echocardiography revealed a perimembranous VSD with left pulmonary artery stenosis. Following birth, he was noted to have persistently elevated serum lactate (4.0–10.0 mmol/l; normal range 0.7–2.2 mmol/l). He exhibited global developmental delay and poor growth with his weight and length remaining below the 3rd centile. Brain MRI at the age of 10 days demonstrated bilateral abnormal high T2 signal intensity of the frontal periventricular area within the regions of caudothalamic grooves and associated irregularity of the outline of the lateral ventricles. Also noted were the foci of abnormal T2 signal in the lentiform nuclei with paucity of periventricular white matter and oedema. The MRI also revealed a Dandy–Walker malformation with partial agenesis of the corpus callosum (Fig EV1C and D). At 5 months of age, there was a rapid increase in head size. Brain CT scan detected acute hydrocephalus with significant dilatation of the third and lateral ventricles. He underwent urgent ventriculoperitoneal shunt insertion. At the age of 19 months, he developed a severe respiratory infection requiring paediatric intensive care unit support and lactate levels were elevated in excess of 20.0 mmol/l (normal range 0.7–2.2 mmol/l). With this intercurrent illness, he

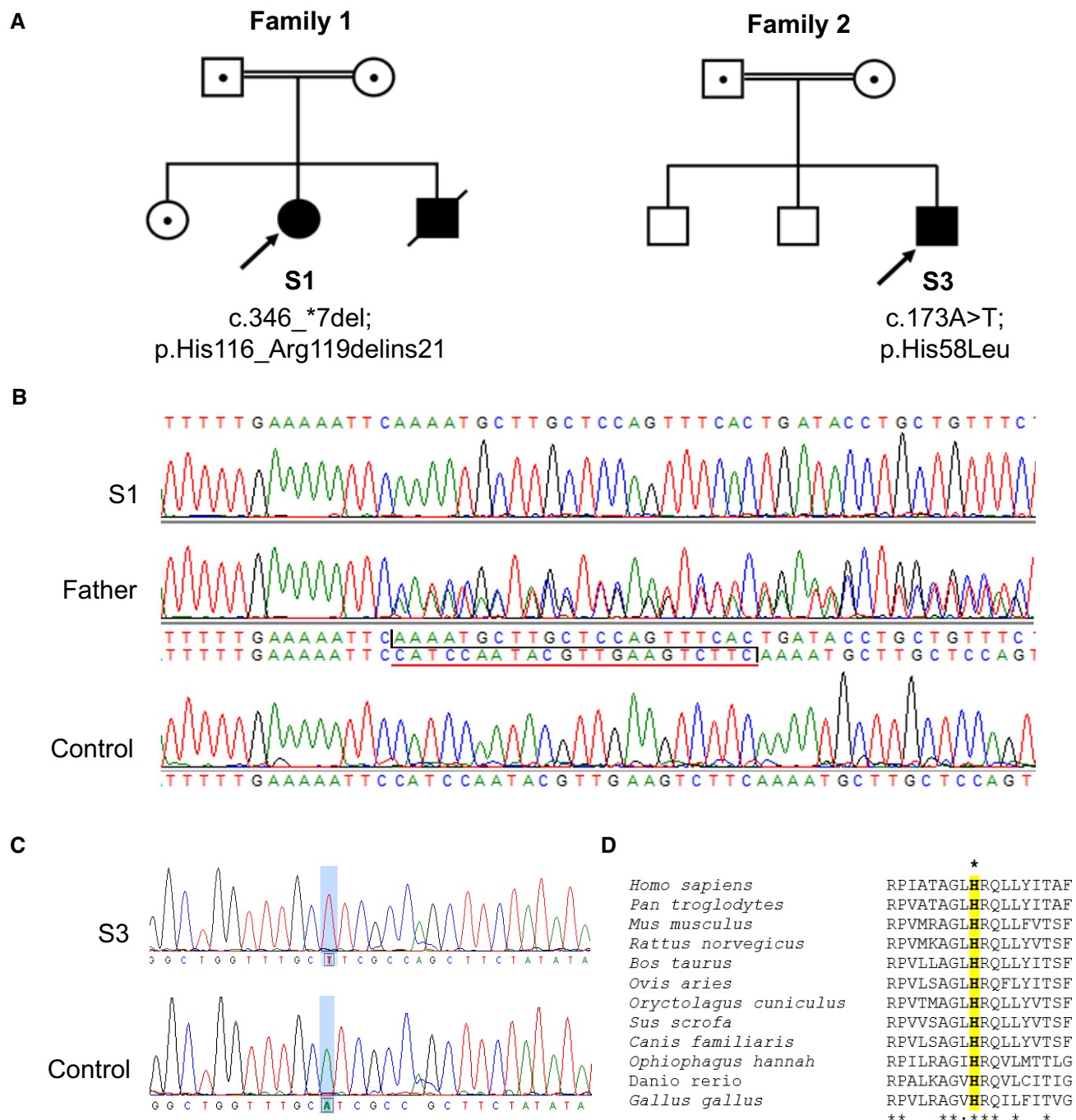

**Figure 1. Genetic analyses of subjects with homozygous *NDUFC2* variants from two unrelated families.**

A  Family pedigrees of Subjects 1, 2 and 3 show segregation of identified homozygous *NDUFC2* variants within each family.

B  Sequencing chromatograms depicting the identified 22 nucleotide deletion (c.346_*7del) identified in Subject 1 (S1), the heterozygous state of the deletion in the father and the wild-type *NDUFC2* sequence in control DNA.

C  Sequencing chromatograms depicting the identified missense variant (c.173A>T; p.(His58Leu)) in Subject 3 (S3) and the wild-type *NDUFC2* sequence in control DNA.

D  Clustal Omega sequence alignment shows evolutionary conservation of the p.His58 residue (highlighted) along with its neighbouring residues (human NDUFC2: NP_004540.1).

developed seizures, regressed and lost all of his previously acquired developmental skills. Recurrent chest infections and pulmonary aspirations suggested an unsafe swallow, and he was placed on nasogastric tube feeding. His clinical course was dominated by progressive spasticity, muscle atrophy, development of a left hydronephrosis and a persistent lactic acidosis. Serum ammonia,

creatine kinase, liver function, thyroid function, plasma amino acids, urinary organic acids and newborn screening results were all unremarkable. He passed away at the age of 3 years as a consequence of pneumonia with associated severe metabolic acidosis.

Subject 3, a male infant, was the third child from healthy consanguineous (first cousin) parents of Moroccan origin with an unremarkable family history (Family 2; Fig 1A). He was born at term after an uneventful pregnancy. Vomiting, failure to thrive, psychomotor delay and poor eye contact were reported from the first months of life. Clinical evaluation performed at 5 months showed impaired growth (weight and length < 3$^{rd}$ centile), microcephaly (< 3$^{rd}$ centile), poor eye contact, non-paralytic strabismus, truncal hypotonia with limb hypertonia, paucity of spontaneous movements and lack of postural control. Brain MRI revealed bilateral abnormal high T2 signal intensity within the thalami (Fig EV1E and F), the basal ganglia and brainstem, though spinal cord MRI was normal. Electroencephalogram (EEG) and peripheral nerve conduction studies were unremarkable, but brainstem auditory and visual evoked potentials showed severe central conduction abnormalities. Epilepsy was reported however, when under anaesthesia to perform MRI, he developed intermittent clonic seizures of both upper limbs which were successfully treated with midazolam. Routine blood investigations including plasma amino acid levels were reported normal, but serum lactate and pyruvate levels were elevated (3.2 mmol/l, normal range 0.8–2.1 mmol/l and 194 μmol/l, normal range 55–145 μmol/l, respectively). Following severe respiratory deterioration, the subject died at 8 months of age.

## Molecular genetic investigations

Initial investigations in Subject 1 focused on commercially available WES, which prioritised known disease genes in the analysis; no candidate variants were identified. Subsequent unbiased WGS identified a homozygous 22-nucleotide deletion within the last exon of the NDUFC2 gene (NM_004549.6). This c.346_*7del NDUFC2 variant (ClinVar: SCV001162791) is predicted to cause a p.(His116_Arg119delins21) stop-loss variant in the NDUFC2 protein (GenBank: NM_004549.6; OMIM: 603845). Segregation analysis using Sanger sequencing found both parents and a healthy brother were heterozygous for the c.346_*7del variant; DNA was not available from Subject 2 (affected sibling) to confirm his genotype (Fig 1A and B). The identified c.346_*7del variant is absent in variant databases, and no healthy controls were homozygous for any loss-of-function NDUFC2 variants within the gnomAD database (Lek et al, 2016).

Identification of an isolated complex I defect in Subject 3′s muscle biopsy prompted genetic studies which identified a homozygous c.173A>T, p.(His58Leu) NDUFC2 variant (ClinVar: SCV001162790) affecting a highly conserved histidine residue (Fig 1C and D). Three healthy controls were heterozygous for a rare variant affecting the same residue (c.172C>T, p.(His58Tyr)) in gnomAD (MAF = 1.21 × 10$^{-5}$). Numerous in silico prediction tools support the pathogenicity of the p.His58Leu residue change [PolyPhen2: 1.000 (probably damaging); SIFT: 0.000 (deleterious); PROVEAN: −6.733 (deleterious); raw CADD: 3.99; scaled CADD: 27.9] (Ng & Henikoff, 2001; Kumar et al, 2009; Adzhubei et al, 2010; Choi et al, 2012; Choi & Chan, 2015; Rentzsch et al, 2018). Carrier testing by Sanger sequencing confirmed that both parents were heterozygous for the c.173A>T variant; the healthy siblings of

Subject 3 were not tested (Fig 1A). These two unrelated families were linked with the aid of GeneMatcher (Sobreira et al, 2015).

Quantitative real-time PCR (qRT–PCR) analysis of fibroblast-derived mRNA transcript levels demonstrated decreased expression of NDUFC2 mRNA in Subject 1 fibroblasts (43% of controls), whilst mRNA levels in Subject 3 fibroblasts were comparable to controls (Fig 2A). Similar investigation of Subject 2 was not possible due to the unavailability of samples; therefore, all further biochemical and functional molecular investigations were performed on samples from Subjects 1 and 3 only.

## Subject fibroblasts and skeletal muscle display isolated complex I deficiency

Assessment of mitochondrial respiratory chain enzyme activities revealed a severe and isolated complex I deficiency in the fibroblasts from Subject 1 (16% residual activity compared to controls) (Fig 2B) and in the fibroblasts (Fig 2B) and muscle (Fig 2C) from Subject 3 (19 and 48% residual activities compared to controls, respectively). High-resolution respirometry revealed both subjects' cell lines exhibited severely decreased rates of oxygen consumption (Fig 2D).

## Subject fibroblasts demonstrate decreased complex I protein levels and complex I assembly

Steady-state protein levels of complex I subunits were assessed by SDS–polyacrylamide gel electrophoresis (SDS–PAGE) of subject and control fibroblast cell lysates. Immunoblot analysis using antibodies against various structural subunits of complex I revealed an overall reduction in the steady-state levels of complex I subunits in both subjects compared to controls (Fig 3A). Assessing the assembly of respiratory chain complexes using blue native polyacrylamide gel electrophoresis (BN-PAGE) of n-dodecyl-β-D-maltoside (DDM)-solubilised mitochondrial preparations from fibroblasts demonstrated undetectable levels of fully assembled complex I in Subject 1 and greatly decreased assembled complex I in Subject 3 compared to controls; all other OXPHOS complexes were unaffected (Fig 3B). BN-PAGE analysis of digitonin-solubilised fibroblast mitochondrial preparations demonstrated a dramatic deficiency of complex I-containing supercomplexes (Fig EV2).

## Expression of wild-type NDUFC2 in patient cell lines using inducible lentiviral transduction ameliorates the complex I assembly defect

Fibroblast cell lines from both Subjects 1 and 3 were transduced with wild-type NDUFC2 using a doxycycline-inducible lentiviral vector to determine whether re-expression of NDUFC2 in the subjects' cells could rescue the phenotype, thus providing further support of variant pathogenicity. Doxycycline induction increased both NDUFC2 levels (and those of complex I subunits NDUFB8, NDUFA9 and NDUFV1) in transduced subject cell lines, compared to the respective uninduced cells and the corresponding untransduced subject cell lines (Fig 3C). In both cases, although the induction of wild-type NDUFC2 protein resulted in an increase of other complex I subunits, it did not restore levels to those observed in controls.

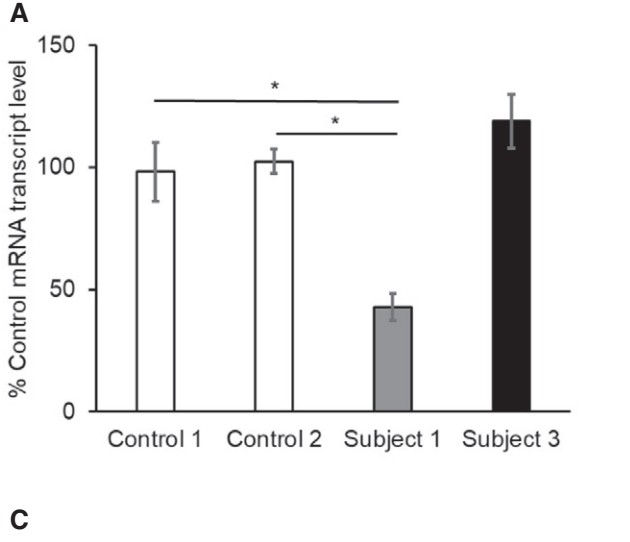

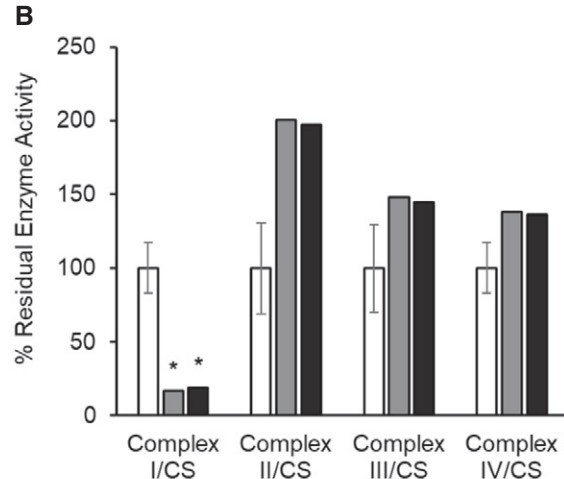

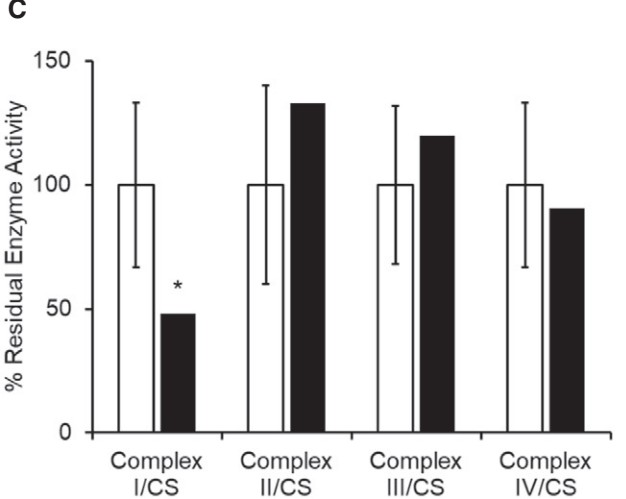

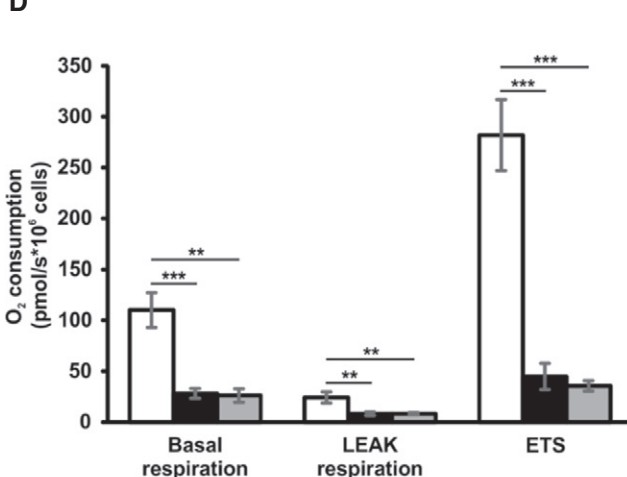

**Figure 2. Biochemical studies and expression of mRNA transcript level in subject fibroblasts.**

A NDUFC2 mRNA transcript levels in fibroblast cell lines of controls (white), Subject 1 (grey) and Subject 3 (black). Quantification of NDUFC2 mRNA transcript levels (normalised to ACTB) showed diminished transcript levels in Subject 1, whilst transcript levels were unaffected in Subject 3. Bars correspond to standard deviations of two independent experiments, performed in duplicate and analysed by Student's t-test; *P < 0.05.

B Assessment of respiratory chain enzyme activities normalised to citrate synthase (CS), in fibroblast cell lines from controls (white), Subject 1 (grey) and Subject 3 (black). Mean residual enzyme activities of control fibroblasts (n = 8) are set at 100%, and error bars represent 1 standard deviation of control activities. Asterisks (*) denote enzyme activity below control range.

C Assessment of respiratory chain enzyme activities normalised to citrate synthase (CS), in skeletal muscle from controls (white) and Subject 3 (black). Mean residual enzyme activities of control skeletal muscle (n > 100 samples) are set at 100%, and bars are 1 standard deviation of control activities. Asterisks (*) denote enzyme activity below control range.

D Respiration of Subject 3 and Subject 1 derived fibroblasts (black and grey, respectively) compared to control fibroblasts (white) using high-resolution respirometry. ROX-corrected analysis of basal respiration, oligomycin-inhibited LEAK respiration and maximum uncoupled respiration (ETS). Both subject fibroblasts exhibited a severe mitochondrial respiration defect as measured by oxygen consumption. Data are mean ± SD from at least 3 experiments and analysed by Student's t-test. **P < 0.01, ***P < 0.001.

Further assessment of complex I assembly in the wild-type NDUFC2 transduced subject fibroblasts showed an increase in complex I assembly with doxycycline induction compared to the respective uninduced cell lines and the respective untransduced subject cell lines (Fig 3D). The increase in assembled complex I reflected the SDS–PAGE findings, where the increase in fully assembled complex I following doxycycline induction was more prominent in the transduced cell line of Subject 3 compared to Subject 1, but the complex I levels in neither cell line reached that of control fibroblasts.

**Complexome profiling confirms stalling in the biogenesis of the complex I holoenzyme**

To gain more insight into the role of NDUFC2 in complex I assembly and stability, we performed complexome profiling using fibroblasts from Subject 1 and Subject 3 and an age-matched control cell line as previously described (Heide et al, 2012; Fuhrmann et al, 2018). In accordance with our previous findings, complex I-containing super-complexes (S) were diminished (Fig 4A–D). Furthermore, there was

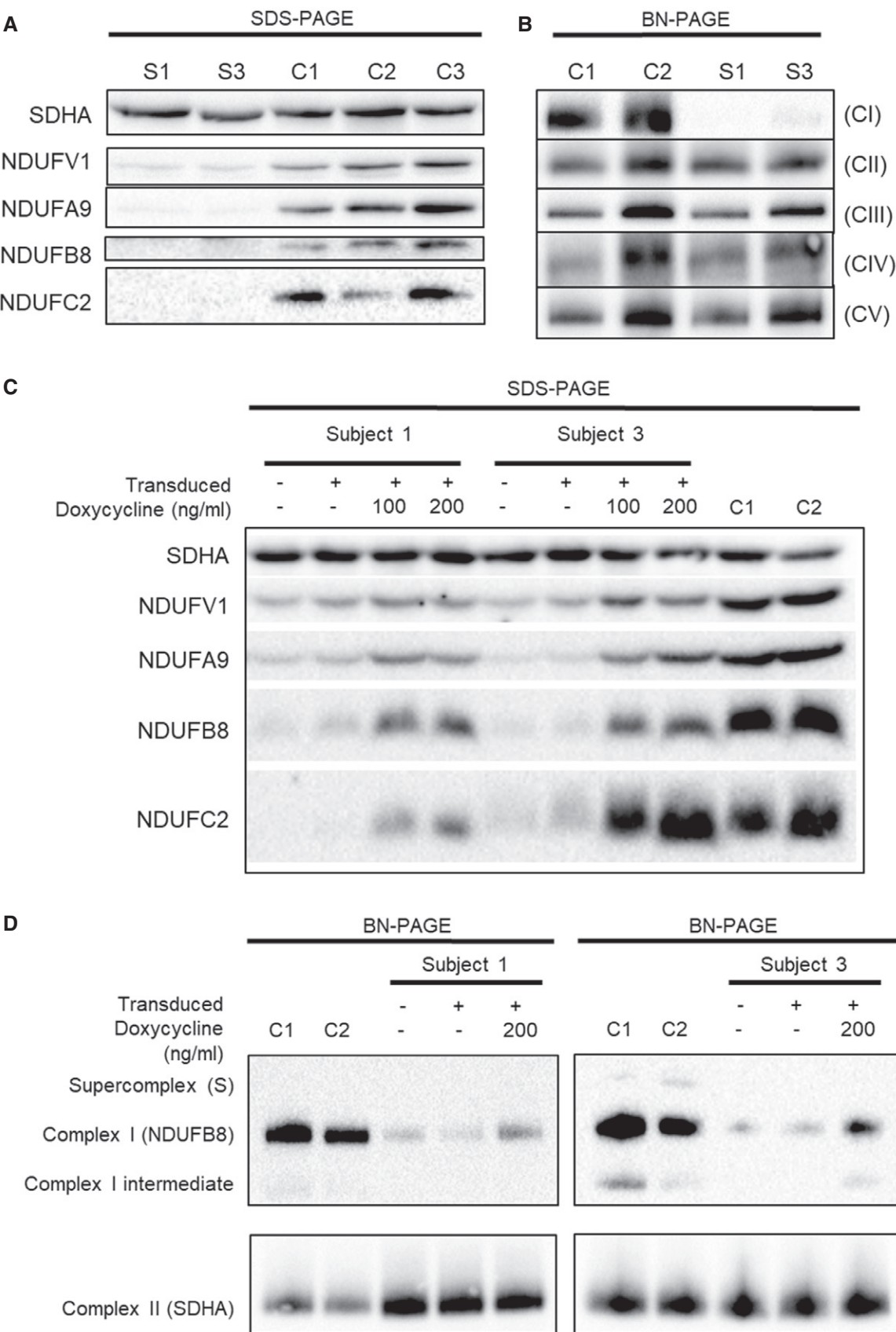

Figure 3.

**Figure 3.  Functional investigation of subject fibroblasts and lentiviral- transduced subject fibroblasts.**

A   Western blot analysis of various structural subunits of complex I in fibroblasts from Subjects 1 and 3 (S1 and S3, respectively) with three age-matched controls (C1, C2 and C3).

B   BN-PAGE analysis of OXPHOS complex assembly in enriched mitochondria from subject and control fibroblasts solubilised using DDM. Immunoblotting was performed using antibodies to structural subunits from each OXPHOS complex [CI (NDUFB8), CII (SDHA), CIII (UQCRC2), CIV (MT-CO1) and CV (ATP5A)].

C   Western blot analysis of whole cell lysates from controls (C1 and C2) and subject cell lines (S1 and S3) that were either untransduced (−) or transduced with the lentiviral vector (pLVX) containing wild-type *NDUFC2* (+). Transduced cell lines were either uninduced or induced with 100 or 200 ng/ml doxycycline (for 48 and 96 h, respectively) where indicated.

D   BN-PAGE analysis of OXPHOS complex assembly in enriched mitochondria isolated from controls (C1 and C2) and subject cell lines that were either untransduced (−) or transduced with the lentiviral vector (pLVX) containing wild-type *NDUFC2* (+). Transduced cell lines were either uninduced or induced with 200 ng/ml doxycycline for 72 h.

Source data are available online for this figure.

no detectable NDUFC2 protein in Subject 1, whilst a scant amount was detectable in Subject 3 (Fig 4A, grey horizontal row); this correlated with the degree of complex I intermediates and supercomplexes detectable in each subject complexome. Abundance of other OXPHOS complexes (complexes III and IV) appeared unaffected compared to control although they remain as individual complexes and smaller supercomplexes lacking complex I (Fig 4C and D, black arrows) rather than forming higher molecular weight supercomplexes (Fig 4A–D). Closer scrutiny of the complex I assembly intermediates in Subject 3 (Fig 4E and F) shows that the NDUFC2 subunit plays an important role in complex I biosynthesis. Assembly of the Q module seems unaffected by the identified variants in both subjects, as Q module subunits (NDUFA5, NDUFS2, NDUFS3, NDUFS7 and NDUFS8) were detected bound to its assembly factors NDUFAF3 and NDUFAF4. In contrast to control, an accumulation of an assembly intermediate comprising the Q module along with the assembly factor TIMMDC1 and the NDUFA13 subunit (Q*~ 300 kDa, Fig 4A, orange dashed boxes; Fig 4E) was identified in Subjects 1 and 3, indicating that this intermediate associates with the membrane. The Q-intermediate with NDUFA13 and TIMMDC1 forms another larger intermediate at 715–800 kDa together with ACAD9, ECSIT, NDUFAF1 and the membrane spanning TMEM126B, all known components of the mitochondrial complex I assembly complex (MCIA). In addition, a preassembled ND4 module together with assembly factors TMEM70 and FOXRED1 was found to accumulate in the subjects' mitochondrial membranes (Fig 4F). Intermediates of the N, ND2 and ND5 module were not detected in fibroblasts from Subjects 1 and 3 (Fig 4A). These findings suggest the stalling of complex I assembly at the Q module formation stage in the absence of the NDUFC2 subunit.

# Discussion

Biochemical investigations into the functional consequences of the identified *NDUFC2* variants showed similar findings for both Subjects 1 and 3, despite one individual harbouring a homozygous missense variant involving a highly conserved residue and the other harbouring a homozygous loss-of-function variant. Similar investigations for Subject 2 were not possible due to a lack of available samples for analysis, but the shared clinical features and *NDUFC2* genotype with Subject 1 (his elder sister) suggest similar biochemical results would have been expected. Review of the gnomAD variant database revealed no homozygous loss-of-function variants within *NDUFC2* and only two homozygous protein-coding variants in healthy controls: p.Leu46Val ($n = 4{,}270$ homozygotes; MAF = 0.197) and p.Arg119His ($n = 1$ homozygote; MAF = $2.77 \times 10^{-4}$); neither substitution was predicted to change the affected residues' biochemical properties by *in silico* prediction tools (Lek *et al*, 2016). Recent work using *NDUFC2*-knockout HEK293T cells showed normal expression of the ND4

**Figure 4.  Complexome Profiling of Fibroblasts from Subjects 1 and 3 confirm complex I deficiency.**

A   Complexome profiling identified assembly intermediates in subject cells with *NDUFC2* variants. Complexes from mitochondrial membranes of control, Subject 1 and Subject 3 fibroblasts were separated by BN-PAGE (Fig EV2) and analysed by complexome profiling. Sample preparation, mass spectrometry, data processing and raw data have been deposited to the ProteomeXchange Consortium via the PRIDE partner repository (Vizcaíno *et al*, 2013) with the dataset identifier ⟨PXD014936⟩. Assignment of complexes: III, complex III; IV, complex IV; III/IV, supercomplex containing dimer of complex III and 1–2 copies of complex IV; S, supercomplex containing complex I, dimer of complex III and 1–4 copies of complex IV. Assembly intermediates are highlighted in dashed boxes indicating stalled complex I assembly. In subject complexomes, various stalled complex I intermediates are assembled: ND4 module (blue dashed box); Q-TIMMDC1-NDUFA13 complex I intermediate (orange dashed box); and Q-TIMMDC1-NDUFA13-MCIA (green dashed box). In control complexome, the Q module (orange dashed box) is assembled without TIMMDC1 or NDUFA13. Complex I modules are denoted according to Stroud *et al* (2016).

B   Abundance data for all the subunits of complex I were averaged to plot an abundance profile of complex I. Heatmap (upper panel) and profile plot (lower panel) compare complex I between control (black), Subject 1 (red) and Subject 3 (blue).

C   Profile plot of complex III (average of all subunits). Black arrows indicate shifted complex III to the position of the individual complex and small supercomplex (III/IV) in Subjects 1 and 3.

D   Profile plot of complex IV (average of all subunits) indicate less complex IV at the position of the supercomplexes for Subject 1 and Subject 3, but higher amounts of individual complex IV and small supercomplex (III/IV, black arrow).

E   Details of Q module-containing assembly intermediates of Subject 3. Intensity-based absolute quantification (iBAQ) values of assembly factors, single subunit and averaged Q module subunits were plotted to show abundance profiles. The cartoon (lower panel) indicates assembled intermediate complexes relative to the plot.

F   Details about ND4 module assembly intermediate of Subject 3. Abundance profile of averaged iBAQ values of ND4 subunits and assembly factors show accumulation of a complex containing the ND4 module, FOXRED1 and TMEM70.

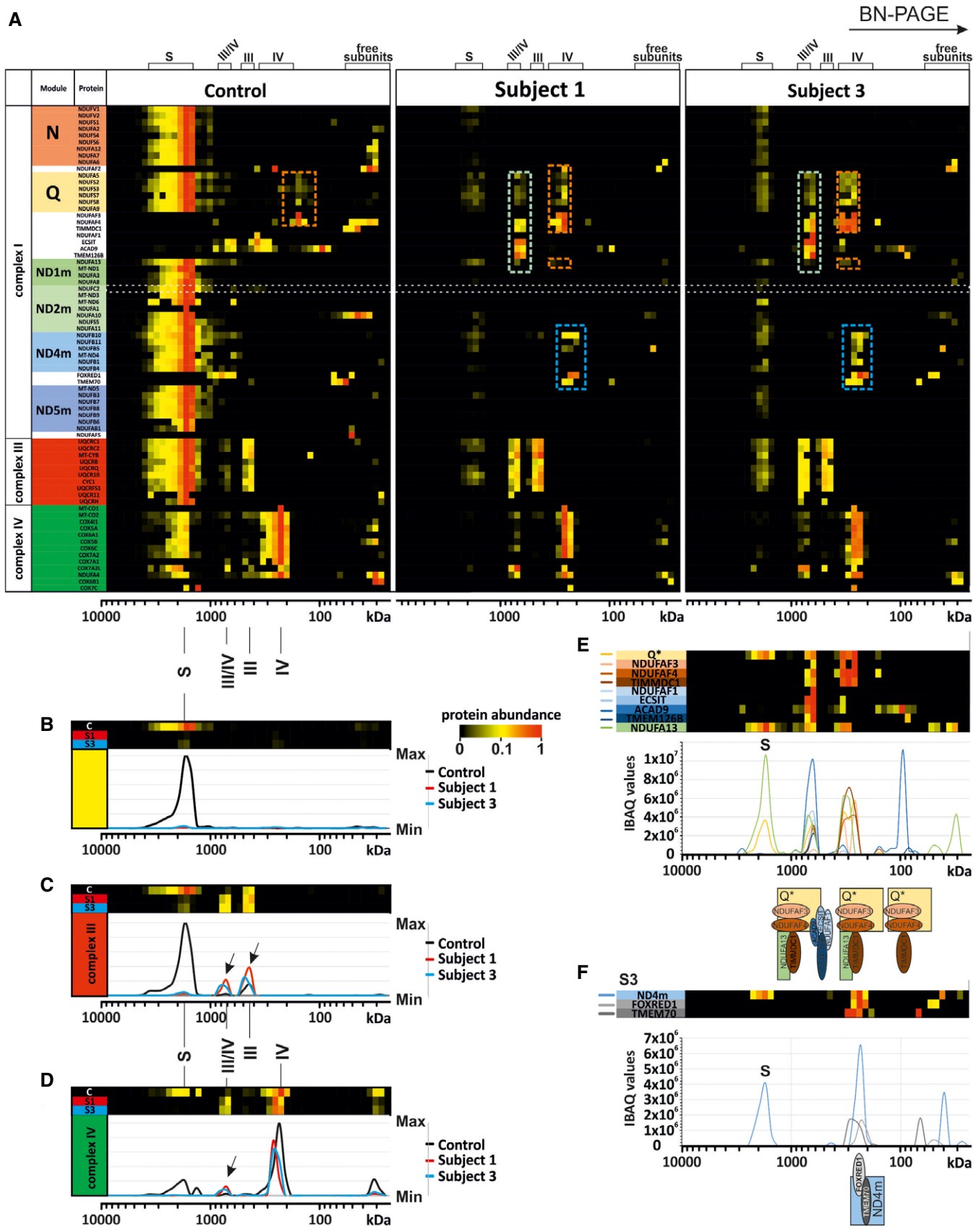

**Figure 4.**

module subunits, but showed a defect in the subunits of the ND2, ND5 and N modules (Stroud *et al*, 2016). Although previous work using *NDUFC2*-knockout HEK293T cells found no effect on the ND1 module subunit levels, our subjects both have defective ND1 module assembly (or stability), with only NDUFA13 detectable. The loss of the NDUFC2-containing ND2 module in Subject 3 asserts that the conserved His58 residue plays an important role in its assembly or stability.

The induction of wild-type NDUFC2 in Subject 1 fibroblasts resulted in increased protein levels, although not as high as the endogenous NDUFC2 level observed in controls. This partial rescue is echoed across the steady-state levels of other complex I subunits and the level of fully assembled complex I. In Subject 3 fibroblasts, NDUFC2 levels following induction of wild-type NDUFC2 appeared similar to endogenous NDUFC2 levels in control fibroblasts, although levels of fully assembled complex I and steady-state levels other complex I subunits still did not reach control levels. It has been suggested that complex I requires just 24 h to fully assemble (Guerrero-Castillo *et al*, 2017) meaning the 72 h of doxycycline induction should provide sufficient time for holoenzyme assembly and incorporation of the induced wild-type NDUFC2 although the presence of mutant NDUFC2 or aberrant subassembly species may affect this process. Despite the levels of protein in the rescued cells

failing to meet endogenous levels in controls, our results clearly demonstrate that lentiviral expression of wild-type *NDUFC2* ameliorates the phenotype, leading not only to increased steady-state levels of complex I subunits, but of their incorporation into the holoenzyme, as demonstrated by increased levels of fully assembled complex I. These results strongly support the pathogenicity of both subjects' *NDUFC2* variants.

According to current knowledge of the assembly sequence of complex I (Guerrero-Castillo *et al*, 2017; Formosa *et al*, 2018; Signes & Fernandez-Vizarra, 2018), the Q-intermediate docks to the ND1 module together with the membrane spanning assembly factor TIMMDC1 to form an intermediate containing the Q module and the ND1 module subunits (Fig 5A). We identified an assembly intermediate in our subjects consisting of the stable Q module in complex with the assembly factors TMEM126B, ACAD9, NDUFAF1 and ECSIT, comprising the mitochondrial complex I intermediate assembly complex (MCIA), TIMMDC1 and (at least) NDUFA13 (Fig 4A, Subject 1 and Subject 3, green dashed boxes, Fig 5C). ND1 and NDUFA8, both previously associated with this stage of assembly, were not detected, but this may reflect the sensitivity of the assay. The presence of the Q module in complex with assembly factors demonstrates that these are indeed assembly

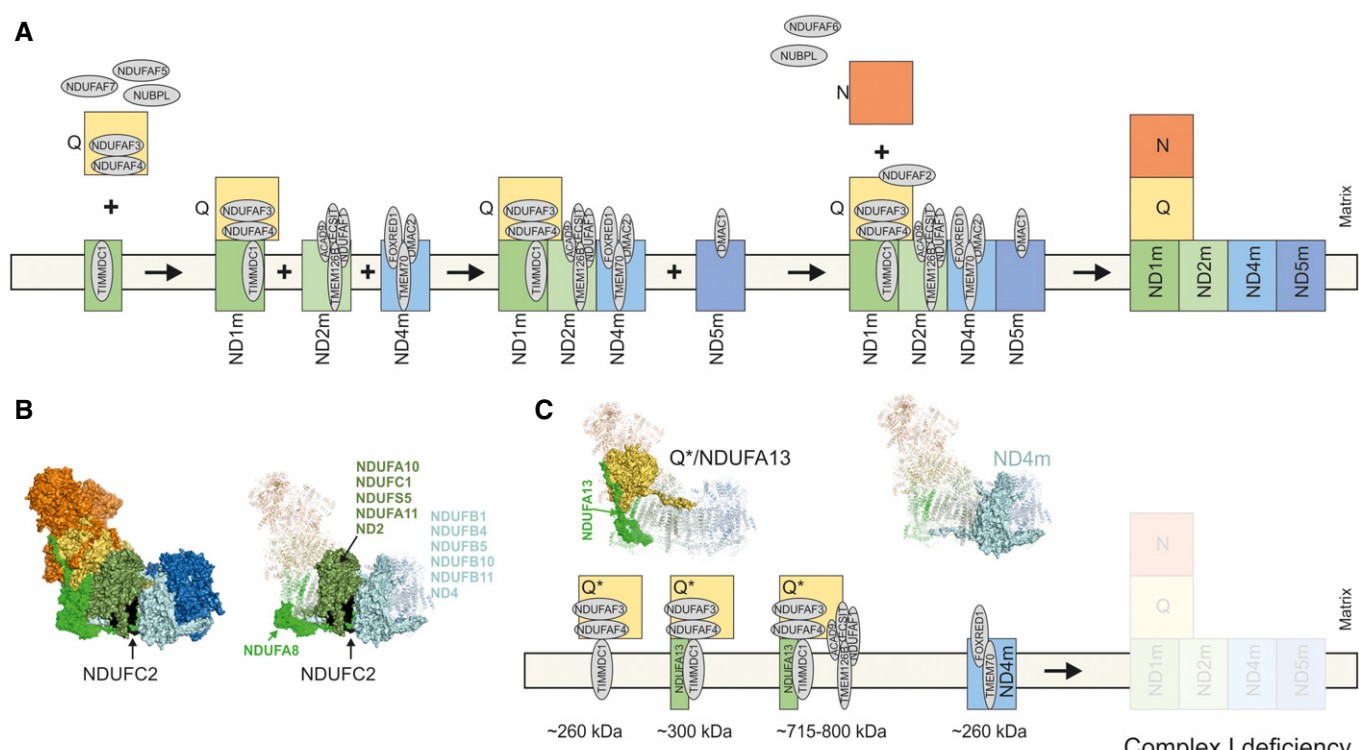

**Figure 5. Molecular consequences of defects in NDUFC2 on complex I assembly.**

A  Assembly sequence of complex I modules according to current state of knowledge in human cells (Guerrero-Castillo *et al*, 2017; Signes & Fernandez-Vizarra, 2018). The ND1 module (ND1 m), ND2 module (ND2m), ND4 module (ND4m) and ND5 module (ND5m) are identified by these abbreviations.

B  Left, structural modules of complex I highlighted in colours based on mouse cryo-EM structure (PDB: 6g2j) (Agip *et al*, 2018). NDUFC2 is shown in black. Right, surface view indicates subunits interacting in a neighbourhood of less than 10 Å to NDUFC2. All other subunits not interacting on any part of the protein are in transparent cartoon view.

C  Assembly intermediates/dead end assembly products in mitochondria from Subject 3 according to data from complexome profiling. Assembled subunits are highlighted in surface view within cryo-EM structure (PDB: 6g2j) (Agip *et al*, 2018).

intermediates rather than degradation products caused by destabilisation of complex I *in situ* during electrophoresis, given any degradation products would not be found associated with complex I assembly factors. NDUFA13 is the sole subunit of ND1 module that is detectable in complex with the Q module within the complexomes of Subjects 1 and 3 (~ 300 kDa, Fig 4A (orange dashed boxes), Figs 4E and 5C). A similar assembly intermediate lacking the MCIA factors was previously observed in cases harbouring pathogenic variants in the MCIA factor TMEM126B where control complexome data also showed a Q module intermediate lacking TIMMDC1, similar to the present control complexome (Alston *et al*, 2016). No other subunit of the ND1 module, nor its neighbouring ND2 module (ND2m), was detected in subject assembly intermediates. The predicted model of complex I assembly implicates the MCIA complex in the assembly and incorporation of the ND2 module (Guerrero-Castillo *et al*, 2017). In the absence of NDUFC2, no ND2 module was detected and an alternative complex I assembly intermediate formed in subject complexomes (~ 715–800 kDa) (Fig 4A, e.g. Subject 1 and Subject 3, green dashed boxes; Fig 4D).

Our molecular data indicate an essential role of NDUFC2 in the assembly of the membrane arm of complex I. NDUFC2 is a membrane protein assigned to the ND2 module within the proton pumping modules (Fig 5B). The protein locates to the intermembrane space and has contact with 12 other subunits (Fig 5B, right panel) including NDUFA8 within the ND1 module, ND2 module subunits (NDUFA10, NDUFA11, NDUFS5, NDUFC1 and ND2) and ND4 module constituents (NDUFB1, NDUFB5, NDUFB10, NDUFB11 and ND4) (Zhu *et al*, 2016; Agip *et al*, 2018). All these contacts may be important for complex I assembly and stability. Complexome profiling of both subjects is consistent with this model, where loss of NDUFC2 causes concurrent loss of its associating subunits—those of the ND1 and ND2 modules. Our data support successful formation of the Q module with subsequent assembly stalling due to failed incorporation of the ND1 subunit into the inner mitochondrial membrane. It is possible that NDUFC2 may represent a scaffold for ND1 module formation; the absence of stable ND1 and ND2 modules indicates that these proximal P-modules do not preassemble independently.

By contrast, the ND4 module, which is comprised of ND4, NDUFB1, NDUFB5, NDUFB10 and NDUFB11 and the assembly factors TMEM70 and FOXRED1, was detected in Subject 3 and to a lesser extent in Subject 1 (~ 260 kDa, Fig 4A, Subject 1 and Subject 3, blue dashed boxes; Figs 4F and 5C). This is consistent with previous assembly models and human complexome data that support independent assembly of the membrane-bound distal P-modules (Alston *et al*, 2016, 2020; Stroud *et al*, 2016; Guerrero-Castillo *et al*, 2017). Neither the ND5 module nor the N module intermediates were identified in the complexome data from the *NDUFC2* subjects. These intermediates are either unstable, turned over by mitochondrial proteases (Szczepanowska *et al*, 2020) or their assembly is sufficiently under the control of middle stage assembly.

Previous reports of gene defects involving the subunits of the ND2 module describe pathogenic variants in *NDUFA1*, *NDUFA10* and *NDUFA11*. Although these all affect the same module, the clinical presentation associated with the different gene defects is highly variable, a common observation in mitochondrial genetic disease. Reported patients harbouring pathogenic variants in *NDUFA1* or *NDUFA10* presented predominantly with Leigh syndrome or with Leigh-like symptoms, whilst the single case of early-onset *NDUFA11*-related mitochondrial disease presented as fatal encephalocardiomyopathy (Fernandez-Moreira *et al*, 2007; Berger *et al*, 2008; Hoefs *et al*, 2010; Minoia *et al*, 2017). A range of molecular, biochemical and functional investigations was performed in these cases, revealing complex I assembly defects (NDUFA1 and NDUFA10) and decreased complex I enzyme activities (NDUFA1, NDUFA10 and NDUFA11). Together, these reports reinforce the importance of ND2 module subunits and the impact of their dysfunction in cases of paediatric mitochondrial disease. Recently, an increasing volume of complexome profiling data has been acquired using cell lines from subjects with genetically confirmed mitochondrial complex I deficiency; some of whom presented with Leigh syndrome (Alston *et al*, 2016, 2018, 2020). This has paved the way to assess and confirm previously deduced steps of complex I assembly. Investigating how pathogenic variants within structural subunits and assembly factors affects complex I assembly revealed novel assembly intermediates that suggest stalling of the assembly process. To date, patient-derived functional investigations have provided data supporting the independent assembly of modules, such as the Q module and ND4 module, and have shed light on the important roles of novel assembly factors including TMEM126B and NDUFAF8. Additionally, they have characterised the functional consequence of pathogenic variants within structural complex I subunits (e.g. NDUFA6) and their effect on the stepwise assembly of the holoenzyme.

In summary, we describe three subjects from independent consanguineous families whose clinical presentations were consistent with a diagnosis of Leigh syndrome and who exhibited radiological lesions of the basal ganglia, thalami and substantia nigra. Genetic analysis identified novel homozygous *NDUFC2* variants in both families, with subsequent biochemical and functional analysis of samples from one affected member of each family demonstrating these variants lead to an isolated complex I enzyme deficiency and a confirmed defect in the assembly of the complex I holoenzyme. Bi-allelic *NDUFC2* variants have not previously been reported in cases of mitochondrial disease; however, defects involving other ND2 module subunits have been associated with complex I deficiency (Fernandez-Moreira *et al*, 2007; Berger *et al*, 2008; Hoefs *et al*, 2010; Minoia *et al*, 2017). Complexome profiling not only confirmed a defect in complex I assembly but also revealed complex I assembly intermediates, providing *in vivo* support of previous findings from *NDUFC2* KO HEK293T cell studies (Stroud *et al*, 2016). Moreover, it indicates an important role for NDUFC2 in the assembly of the membrane arm of complex I, particularly involving the ND2 module and, possibly, the ND1 module. Complexome profiling of complex I-deficient subject cell lines continues to be a useful tool in elucidating the molecular mechanisms underlying complex I assembly and stability (Szczepanowska *et al*, 2020).

# Materials and Methods

All experiments conformed to the principles set out in the Department of Health and Human Services Belmont Report.

### Genetic analysis

Commercial NGS services provided by CentoGene (Germany) were used to perform WES analysis on DNA from Subject 1.

Genetic studies were performed on Subject 3 using a custom 300 gene targeted gene panel of nuclear genes previously associated with mitochondrial disease, candidate genes that take part in the same molecular pathway, and included all genes encoding complex I subunits. Library capture and enrichment were performed using a custom panel (SureSelectXT Custom library, Agilent, US), and sequencing was performed on an Illumina MiSeq platform according to the manufacturer's protocol. Bioinformatic analysis was performed as previously described (Legati *et al*, 2016).

Informed consent for diagnostic and research studies was obtained for all subjects in accordance with the Declaration of Helsinki protocols and approved by local institutional review boards.

RNA purification and cDNA retrotranscription were performed using RNeasy mini kit (QIAGEN) and GoTaq 2-Step RT-qPCR System (Promega), respectively, according to the manufacturers' protocols. RNA was extracted from skin fibroblasts, and 1 μg was used as template for RT–PCR, with the following oligonucleotides sequences: q-NDUFC2-F 232_251 (CTGCTCCGGCCTGATTGATA) and q-NDUFC2-R 358_377 (GGTCCCTCACAGCATACAGG) for *NDUFC2* (NM_004549.6); q-ACTB-F 425_442 (CCAACCGCGAGAA GATGA) and q-ACTB-R 502_521 (CCAGAGGCGTACAGGGATAG) for *ACTB* (NM_001101.5), used for normalisation.

## Cell culture

Skin fibroblast was cultured in medium containing DMEM (supplemented with 1 mM pyruvate, 4.5 g/l glucose) (Thermo Fisher Scientific), 10% foetal calf serum (Thermo Fisher Scientific), 50 μg/ml uridine (Sigma), 1× non-essential amino acids (Thermo Fisher Scientific) and 1% penicillin/streptomycin (Thermo Fisher Scientific) at 37°C in 5% $CO_2$. During the lentiviral rescue experiment, skin fibroblasts were cultured in the presence of dialysed foetal calf serum (Thermo Fisher Scientific).

## Oxygen consumption

Mitochondrial respiration of intact cells was measured using high-resolution respirometry (Oxygraph-2k, Oroboros Instruments, Innsbruck, Austria) with DatLab software 7.1.0.21 (Oroboros Instruments, Innsbruck, Austria) (Pesta & Gnaiger, 2012). Measurements were performed at 37°C in full growth medium (DMEM, 10% foetal calf serum, 1 mM pyruvate, 4.5 g/l glucose, 50 μg/ml uridine, 1× non-essential amino acids, 1% penicillin/streptomycin). Basal respiration was measured for 20 min followed by titration of oligomycin (2.5 μM final concentration; f.c.) to measure oligomycin-inhibited LEAK respiration. Subsequently, the respiratory uncoupler FCCP (Sigma-Aldrich, Munich, Germany) was titrated stepwise until maximal uncoupled respiration (ETS) was reached. Residual oxygen consumption (ROX) was determined after sequential inhibition of complex I with rotenone and complex III with antimycin A. Absolute respiration rates were corrected for ROX and normalised for cell number.

## Respiratory chain enzyme activities

Mitochondrial respiratory chain enzyme activities were assayed spectrophotometrically as previously described in subject fibroblasts (Frazier *et al*, 2020) and in muscle samples (Bugiani *et al*, 2004).

## Blue native electrophoresis and in-gel activity stains

Sample preparation and BN-PAGE of cultured cell pellets were essentially performed as previously described (Wittig *et al*, 2006). Equal protein amounts of samples were loaded on top of a 3–18% acrylamide gradient gel (dimension 14 × 14 cm). After native electrophoresis in a cold chamber, blue native gels were fixed in 50% (*v*/*v*) methanol, 10% (*v*/*v*) acetic acid and 10 mM ammonium acetate for 30 min and stained with Coomassie (0.025% Serva Blue G, 10% (*v*/*v*) acetic acid). In-gel activity stains were performed as described in Wittig *et al* (2007).

## Western blotting and Blue Native-PAGE

Steady-state protein levels of complex I subunits were assessed by SDS–polyacrylamide gel electrophoresis (SDS–PAGE), and OXPHOS complexes were assessed in enriched mitochondrial pellets solubilised with DDM (Sigma-Aldrich) by BN-PAGE in subject and control fibroblast cells as previously described (Oláhová *et al*, 2015).

Immunoblotting was carried out using primary antibodies against various OXPHOS subunits all used at a dilution of 1 in 1,000 [NDUFC2 (Abcam ab192265), NDUFB8 (Abcam ab110242), NDUFA9 (Abcam ab14713), NDUFV1 (Proteintech 11238-1-AP), SDHA (Abcam ab14715), UQCRC2 (Abcam ab14745), MT-CO1 (Abcam ab14705) and ATP5A (Abcam ab14748)] followed by species appropriate HRP-conjugated secondary antibodies (Dako: anti-rabbit (P0399) 1 in 3,000 dilution and anti-mouse (P0260) 1 in 2,000 dilution).

## Complexome profiling

Complexome profiling was performed on fibroblasts from Subject 1 and Subject 3 and an age-matched control cell line as previously described (Heide *et al*, 2012; Fuhrmann *et al*, 2018).

Sample preparation, mass spectrometry, data processing and raw data have been deposited to the ProteomeXchange Consortium http://proteomecentral.proteomexchange.org) via the PRIDE partner repository (Vizcaíno *et al*, 2013) with the dataset identifier (http://proteomecentral.proteomexchange.org/cgi/GetDataset?ID = PXD014936). Complexome data were further analysed using NOVA (Giese *et al*, 2015). Complex I and identified assembly intermediates were visualised using the EM structure of murine complex I (PDB: 6g2j) (Agip *et al*, 2018) and PyMOL (The PyMOL Molecular Graphics System, version 2.3.3. Schrödinger, LLC.).

## Lentiviral rescue

Wild-type *NDUFC2* was expressed in patient fibroblasts using the Lenti-X TetOne Inducible Expression System. Wild-type *NDUFC2* insert was generated by PCR using cDNA from control fibroblasts and the following primers containing BamHI and EcoRI restriction sites, respectively: Forward: 5′-GAGGTGGTCTGGATCCTCAACG TA TTGGATGGAATTTTTCAAAAATTTCA-3′; Reverse: 5′-CCCTCGTAA AGAATTCATGATCGCACGGCGGAACCCA-3′. The insert was cloned into the linearised pLVX-TetOne-Puro plasmid vector (Takara Bio/ Clontech) using the In-Fusion HD Cloning Kit (Takara Bio/Clontech). The *NDUFC2* containing vector was packaged into infectious lentiviral particles using Lenti-X Packaging Single Shots (Takara

Bio/Clontech) to transfect HEK293T according to manufacturer's guidelines. Media containing infectious lentivirus was harvested after 48 h and centrifuged at 500 *g* for 10 min to remove cellular debris, and the supernatant was retained. This virus containing supernatant was used to transduce Subject fibroblast cell lines at a ratio of 1:2 with fresh media and 4 mg/ml polybrene overnight. The media was replaced and puromycin (2 µg/ml) added to select for successfully transduced cells. Transduced cells were cultured in puromycin for 7 days before being maintained in standard DMEM. Expression of wild-type *NDUFC2* was induced by adding doxycycline (Takara Bio/Clontech; 100–200 µg/ml) to the media and incubating for 48–96 h.

## Data availability

The datasets produced in this study are available in the following database: Complexome profiling data: ProteomeXchange consortium via the PRIDE repository, dataset identifier PXD014936 (http://www.ebi.ac.uk/pride/archive/projects/PXD014936).

**Expanded View** for this article is available online.

## Acknowledgements

Work in our laboratories is supported by the Wellcome Centre for Mitochondrial Research (203105/Z/16/Z), the Medical Research Council (MRC) International Centre for Genomic Medicine in Neuromuscular Disease, Newcastle University Centre for Ageing and Vitality [supported by the Biotechnology and Biological Sciences Research Council and Medical Research Council (G016354/1)], the UK NIHR Biomedical Research Centre in Age and Age Related Diseases award to the Newcastle upon Tyne Hospitals NHS Foundation, the MRC/ESPRC Newcastle Molecular Pathology Node, the UK National Health Service Highly Specialised Service for Rare Mitochondrial Disorders, the Lily Foundation, the Pierfranco and Luisa Mariani Foundation, the E-Rare project GENOMIT and the Italian Ministry of Health (grant GR-2016-02361241). AA holds a PhD studentship funded by the Kuwait Civil Services Commission. CLA is supported by the National Institute for Health Research (NIHR Post-Doctoral Fellowship, PDF-2018-11-ST2-021). IW is supported by the Deutsche Forschungsgemeinschaft: (SFB 815/Z1 and EXC2026: Cardio Pulmonary Institute (CPI)) and by the BMBF mitoNET—German Network for Mitochondrial Disorders 01GM1906D. We thank the "Cell line and DNA Bank of Genetic Movement Disorders and Mitochondrial Diseases" of the Telethon Network of Genetic Biobanks (grant GTB12001J) and Eurobiobank Network which supplied biological specimens. The authors would like to thank both families who participated in this study. The views expressed in this publication are those of the author(s) and not necessarily those of the NHS, the National Institute for Health Research or the Department of Health and Social Care.

## Author contributions

IW, DG and RWT conceptualized and designed the study. AAla, AN, JH, KT, MO, AL, EL, JM, MS, LH, SA, FH, AAlm, AAr, CLA, RM and IW acquired and analysed the data. AAla, KT, IW and RWT drafted the manuscript. AN, JH, MO, AL, EL, JM, MS, LH, SA, FH, AAlm, AAr, CLA, RM and DG critically revised the manuscript. RM, IW, DG and RWT supervised the study.

## Conflict of interest

The authors declare that they have no conflict of interest.

## The paper explained

### Problem

Mitochondrial disease is the umbrella term for a group of disorders characterised by mitochondrial dysfunction, predominantly resulting in decreased ATP production by the oxidative phosphorylation (OXPHOS) system. These exhibit huge variability, not only in terms of their clinical manifestations, but also the underlying genetic basis. Mitochondrial disease can be caused by changes within the mitochondrial genome itself (mtDNA) or changes in nuclear genes that encode one of ~ 1,200 proteins that function in mitochondria. Of these, more than 300 have already been associated with mitochondrial disease and this diversity can make obtaining genetic diagnoses in affected individuals difficult. NDUFC2 is a mitochondrial protein encoded by the nuclear genome and is a subunit of mitochondrial complex I, one of 5 multisubunit complexes that comprise the OXPHOS system. To date, there have not been any previous reports of disease-causing *NDUFC2* variants in cases of mitochondrial disease.

### Results

We used whole exome and whole genome sequencing to identify recessively inherited variants in *NDUFC2* (NM_004549.6; c.346_*7del, p.(His116_Arg119delins21)) in one family with two affected siblings and a different recessively inherited variant (c.173A>T, p.(His58Leu)) in an affected individual from an unrelated family. All affected individuals presented with a clinical picture consistent with Leigh Syndrome, a mitochondrial disorder that has many possible genetic causes, including developmental delay, lactic acidosis and characteristic neuroimaging abnormalities. We showed that mitochondrial complex I function was impaired in patient cells and muscle samples and crucially showed an amelioration of this deficiency when introducing a wild-type (normal) copy of *NDUFC2* using a lentiviral delivery system. This functional rescue confirms the identified *NDUFC2* gene variants are disease-causing. Furthermore, complexome analysis of fibroblast mitochondria showed a loss of fully assembled complex I and an accumulation of specific assembly intermediates in the patients compared to controls, suggesting a key role for NDUFC2 in complex I biogenesis.

### Impact

Our study documents the first cases of mitochondrial disease caused by variants in the *NDUFC2* gene. Confirmation of these variants as causative enables prenatal testing of a subsequent pregnancies in these families, demonstrating the clinical importance of providing a genetic diagnosis in mitochondrial disease cases; in addition, *NDUFC2* may now be screened as a candidate gene in genetically undiagnosed cases of Leigh syndrome. Our work also provides clearer insight into the function of NDUFC2 through a role in the assembly of the membrane arm of complex I, specifically the ND2 module.

## For more information

(i) OMIM NDUFC2 page: https://www.omim.org/entry/603845?search=NDUFC2&highlight=ndufc2.

(ii) Author's website: https://www.newcastle-mitochondria.com.

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
