## [Review Process File · EMBO Molecular Medicine]

BI-ALLELIC PATHOGENIC VARIANTS IN NDUFC2 CAUSE EARLY-ONSET LEIGH SYNDROME AND STALLED BIOGENESIS OF COMPLEX I

Ahmad Alahmad, Alessia Nasca, Juliana Heidler, Kyle Thompson, Monika Olahova, Andrea Legati, Eleonara Lamantea, Jana Meisterknecht, Manuela Spagnolo, Langping He, Seham Alameer, Fahad Hakami, Abeer Almehdar, Anna Ardisson, Charlotte Alston, Robert McFarland, Ilka Wittig, and Daniele Ghezzi, and Robert Taylor

DOI: [10.15252/emmm.202012619](https://doi.org/10.15252/emmm.202012619)

Corresponding author: Robert Taylor (robert.taylor@ncl.ac.uk)

Review Timeline:

Submission Date:	29th Apr 20
Editorial Decision:	17th Jun 20
Revision Received:	24th Jul 20
Editorial Decision:	4th Aug 20
Revision Received:	14th Aug 20
Accepted:	26th Aug 20

Editor: Zeljko Durdevic

Transaction Report:

17th Jun 2020

Dear Prof. Taylor,

Thank you for the submission of your manuscript to EMBO Molecular Medicine, and please accept my apologies for the delay in getting back to you. We have received feedback from two of the three reviewers who agreed to evaluate your manuscript. Should referee #3 provide a report, we will send it to you, with the understanding that we will not ask for an additional revision. As you will see from the reports below, both referees are positive and find the study interesting and important. However, they also have a few suggestions and some minor criticisms that I would like you to address in a revision of the current manuscript. Particular attention should be given to the lack of data from Subject 2 as indicated by both referees. This should be addressed either by performing key experiments in Subject 2 fibroblasts or by providing a rationale for using only Subjects 1 and 3. If the experimentation is not possible due to the lack of material from the deceased Subject 2 it should be clearly stated in the manuscript text that only 2 out of 3 Subjects were investigated in detail.

Addressing the reviewers' concerns in full will be necessary for further considering the manuscript in our journal. Please note that EMBO Molecular Medicine encourages a single round of revision only and therefore, acceptance or rejection of the manuscript will depend on the completeness of your responses included in the next, final version of the manuscript. For this reason, and to save you from any frustrations in the end, I would strongly advise against returning an incomplete revision.

We would welcome the submission of a revised version within three months for further consideration. However, we realize that the current situation is exceptional on the account of the COVID-19/SARS-CoV-2 pandemic. Please let us know if you require longer to complete the revision.

I look forward to receiving your revised manuscript.

***** Reviewer's comments *****

Referee #1 (Remarks for Author):

This manuscript by the group of Taylor presents for the first time patients with mutation in the accessory complex I subunit gene NDUFC2. It is easy to follow and contributes an important piece of information to the field of mitochondrial disease and complex I assembly. However, there are some points that require explanation as listed below.

Major points

In my set of the manuscript there was no figure 2A containing qRT-PCR data (as stated on page 9; I guess this is Fig. 2D?). This should be addressed. Also, for completeness, the qRT-PCR data for Subject 2 should be included here. I guess due to the omission of Fig. 2A, the results section on figure 2 refers to the wrong panels.

Please report (e.g. in a table) also the respiratory chain enzyme activities normalized on mg protein. Normalization on CS can only be used if CS activities do not differ between samples.

What is exactly the reason for not including Subject 2 in figure 3A-B? Since this data was obtained in fibroblasts, I guess these cells are still available for analysis?

Since complexome profiling involves BN-PAGE, CI destabilization in situ might manifest itself as CI falling apart on the BN-PAGE gel. In this sense the latter technique (also) reports on CI stability and not only on CI assembly. This is illustrated by the fact that Subject 1 does not contain fully assembled

CI but still displays non-zero CI catalytic activity. This suggests that in situ CI still has activity and therefore has to be in an assembled state? The same remark could be made about analysis of supercomplexes by BN-PAGE. The above should at least be discussed, as well the potential impact of this "problem" on the obtained results.

Please provide detailed information (if applicable) on whether the presented blots and images were processed and how this was done. E.g. contrast enhancement etc.

Minor:

The authors could refrain from stating on several places in the manuscript that they describe 3 subjects since they investigated only 2 of them in detail.

Introduction, P4: it states: "As a result,..." why are the different modes of inheritance a result from 330 genes being associated with mitochondrial disorders? In my view this sentence is not logical.

Regarding the incomplete rescue in the doxycyclin model. Could it be that due to different genomic integration between cells not all cells (similarly) express the non-mutated NDUFC2 protein upon induction?

Referee #2 (Remarks for Author):

Authors describe two families with biallelic variants in NDUFC2 gene leading to an isolated complex I enzyme deficiency and a confirmed defect in the assembly of the complex I holoenzyme. One individual from family 1 harboring a homozygous missense variant involving a highly conserved residue and second individual from family 2 a homozygous loss-of-function variant. Authors have performed sufficient amount of functional studies to prove the pathogenicity of these variants in NDUFC2 gene. There is one weak point, authors have not investigated individual S2 in Family 1. But I understand that it is always not possible to get the material from deceased individuals.

I have some minor corrections:

I suggest to use word "variant" or pathogenic variant" instead of mutation in the title. See section Terminology:

https://www.acmg.net/docs/Standards_Guidelines_for_the_Interpretation_of_Sequence_Variants.pdf

Introduction:

Please write out abbreviation CSF

Page 6, Case reports:

I suggest to write out "Supplementary figure 1a and 1b, etc or Supp figure" throw-out the manuscript. The abbreviation "Figure EV" is confusing. Or please give an explanation for EV.

Please add to Family 1 and 2 the citation to Figure 1a - pedigrees of both families.

Page 8, Molecular genetic investigations:

Para: "Quantitative real-time PCR (qRT-PCR) analysis of fibroblast-derived mRNA transcript levels demonstrated decreased expression of NDUFC2 mRNA in Subject 1 fibroblasts (43% of controls), while mRNA levels in Subject 3 fibroblasts were comparable to controls (Figure 2A)."

I do not understand how Figure 2A express these results? Please correct "Figure 2D".

Page 9, Subject fibroblasts and skeletal muscle display isolated complex I deficiency:

Here is the same. The citation of Figure 2 subunits is not right. Please correct.

EMM-2020-12619, Alahmad *et al.*
Response to Reviewer's comments

Referee #1 (Remarks for Author):

This manuscript by the group of Taylor presents for the first time patients with mutation in the accessory complex I subunit gene *NDUFC2*. It is easy to follow and contributes an important piece of information to the field of mitochondrial disease and complex I assembly. However, there are some points that require explanation as listed below.

Author's Response: We would like to thank the reviewer for their comments regarding the importance of this work and we will endeavour to address each of the concerns listed below in turn.

Major points:

In my set of the manuscript there was no figure 2A containing qRT-PCR data (as stated on page 9; I guess this is Fig. 2D?). This should be addressed. Also, for completeness, the qRT-PCR data for Subject 2 should be included here. I guess due to the omission of Fig. 2A, the results section on figure 2 refers to the wrong panels.

Author's Response: We would like to thank the reviewer for pointing out the incorrect referencing of Figure 2, we have now rearranged the Figure panels so that Fig 2A is in fact the qRT-PCR data. Regarding the omission of data for Subject 2, this is due to a lack of any available samples from this deceased individual. This is a point brought up again later and something raised by the other reviewer so we have made amendments to the text to make it clear that whilst we report the clinical details of three subjects, only two of these were studied in detail due to the availability of biological samples.

Page 9: added text: "Similar investigation of Subject 2 was not possible due to the unavailability of samples, therefore all further biochemical and functional molecular investigations were performed on samples from Subjects 1 and 3 only."

Please report (e.g. in a table) also the respiratory chain enzyme activities normalized on mg protein. Normalization on CS can only be used if CS activities do not differ between samples.

Author's Response: While we acknowledge that normalizing to total protein is another way to present the respiratory chain data, we respectfully disagree with the reviewer that it would be superior to normalizing to citrate synthase (CS) activity. CS activity is a validated biomarker of mitochondrial content in skeletal muscle and we believe this is in fact more reliable than using total protein content which could potentially be more variable due to the quality of the muscle samples themselves; indeed it is widely used by many diagnostic laboratories offering these bespoke assays across the world (<https://pubmed.ncbi.nlm.nih.gov/32183956/>). In this particular case either normalisation method yields almost identical results - normalization to CS of the CI activity in Subject 3's muscle yields a figure of 48% residual CI activity, whereas normalizing to total protein gives a 49% residual CI activity. On this basis, we feel the data are appropriately described and graphically represented within Figure 2B-C without the need for a further table.

What is exactly the reason for not including Subject 2 in figure 3A-B? Since this data was obtained in fibroblasts, I guess these cells are still available for analysis?

Author's Response: As clarified above, Subject 2 was not included in the functional workup due to the lack of available biological samples, particularly a primary fibroblast cell line. The clinical description is included as we believe this has value due to the similarity in presentation to his older sibling (Subject 1). In the 'Molecular Genetic Investigations' section we state that the genotype of

Subject 2 could not be confirmed due to a lack of any samples, but we accept that we need to make this point clearer throughout the manuscript, so we have made the following changes:

Page 5: "This report highlights the clinical, biochemical and molecular findings of three paediatric subjects from two unrelated consanguineous families" has been changed to "This report highlights the clinical findings of three paediatric subjects from two unrelated consanguineous families".

Page 9: Relating to the qRT-PCR data we have now added "Similar investigation of Subject 2 was not possible due to the unavailability of samples, therefore all further biochemical and functional molecular investigations were performed on samples from Subjects 1 and 3 only."

Page 12: We have added the following point to the discussion; "Similar investigations for Subject 2 were not possible due to a lack of available samples for analysis, but the shared clinical features and *NDUFC2* genotype with Subject 1 (his elder sister) suggests similar biochemical results would have been expected."

Page 15: In the summary we have changed "Subsequent genetic, biochemical and functional analyses have identified novel homozygous *NDUFC2* variants leading to an isolated complex I enzyme deficiency and a confirmed defect in the assembly of the complex I holoenzyme." to now read "Genetic analysis identified novel homozygous *NDUFC2* variants in all three cases, with subsequent biochemical and functional analysis of samples from one affected member of each family demonstrating these variants lead to an isolated complex I enzyme deficiency and a confirmed defect in the assembly of the complex I holoenzyme."

Since complexome profiling involves BN-PAGE, CI destabilization in situ might manifest itself as CI falling apart on the BN-PAGE gel. In this sense the latter technique (also) reports on CI stability and not only on CI assembly. This is illustrated by the fact that Subject 1 does not contain fully assembled CI but still displays non-zero CI catalytic activity. This suggests that in situ CI still has activity and therefore has to be in an assembled state? The same remark could be made about analysis of supercomplexes by BN-PAGE. The above should at least be discussed, as well the potential impact of this "problem" on the obtained results.

Author's Response: We thank the reviewer for their comments, we acknowledge that it can be difficult to differentiate assembly defects with destabilisation of assembled complexes and supercomplexes. In this case, we do see a very small amount of CI-containing supercomplexes and some very weak staining in the in gel (NADH-reductase) assay in Subjects 1 and 3 (see **Figure EV2**), which likely accounts for the residual CI activity detected. The key reason we believe this is primarily a problem with assembly rather than stability is that the assembly intermediates detected correspond to those currently described in the literature and, most importantly, the observed intermediates are found with assembly factors bound which would not be the case if fully assembled CI was destabilised during electrophoresis. We have added the following text to highlight this in the discussion:

Page 13: "The presence of the Q module in complex with assembly factors demonstrates that these are indeed assembly intermediates rather than degradation products caused by destabilisation of complex I in situ during electrophoresis, given any degradation products would not be found associated with complex I assembly factors."

Please provide detailed information (if applicable) on whether the presented blots and images were processed and how this was done. E.g. contrast enhancement etc.

Author's Response: The presented western blot images were taken with a CCD camera (Bio-Rad Gel Doc) detecting a chemiluminescent signal after incubating the membranes with ECL prime (Amersham). Multiple exposures were taken for each membrane to get appropriate images for the various proteins detected as antibody quality and expression levels of each individual protein leads to variation in signal strength. The ImageLab software used can detect saturated pixels so only

exposures without any saturation were used for producing the figures. The contrast was set with the auto setting in the software on a non-saturated image and was not manually altered for any individual panels.

Minor points:

The authors could refrain from stating on several places in the manuscript that they describe 3 subjects since they investigated only 2 of them in detail.

Author's Response: As stated above, we have now amended the manuscript text to clarify this point. We do provide a clinical description of all three subjects from the two families, but recognise that our biochemical investigations are limited to two individuals due to a lack of biological samples from Subject 2.

Introduction, P4: it states: "As a result,..." why are the different modes of inheritance a result from 330 genes being associated with mitochondrial disorders? In my view this sentence is not logical.

Author's Response: We agree that this could be made clearer. The wide array of potential genetic causes and the fact that mutations in either the mitochondrial or nuclear genome can lead to mitochondrial disease is the reason these disorders can be passed down through any mode of inheritance. We have amended this sentence and the one immediately preceding to now read:

Page 4: "Mitochondrial proteins are encoded by either the nuclear genome (involving >1000 genes) or the mitochondrial genome (13 protein-coding genes), with pathogenic variants in more than 330 genes having been associated with mitochondrial disorders to date (Thompson *et al*, 2019). Due to this genetic heterogeneity, mitochondrial disease can follow any mode of inheritance including maternal, autosomal (dominant or recessive) or X-linked inheritance or occur *de novo*"

Regarding the incomplete rescue in the doxycycline model. Could it be that due to different genomic integration between cells not all cells (similarly) express the non-mutated *NDUFC2* protein upon induction?

Author's Response: Yes, we agree with the reviewer that this could be part of the reason, although the cells were transduced at a low multiplicity of infection (MOI) and were selected using puromycin. At most, 30% of the cells were puromycin resistant and had therefore been successfully transduced with the lentiviral vector. This suggests that the vast majority of cells will have only taken up a single copy of the vector. Whilst the genetic integration will not have been in the same location in each cell, the wild type *NDUFC2* should be as a single copy and is under the same inducible promoter in each cell so we would expect similar expression levels in each cell. This is supported by the fact that we carried out multiple transductions independently and did not observe a noticeable difference in *NDUFC2* expression between each population of transduced cells. It is worth noting that these were populations of cells that were transduced in a single dish, subjected to puromycin selection and then propagated into multiple flasks for the uninduced and various induced concentrations tested. They were not grown up from single colonies of transduced fibroblasts as these were primary fibroblasts and growing enough cells for use in the experiments from a single transduced fibroblast cell was not possible due to the number of rounds of cell division required. We acknowledge that some individual clones may well have had better expression than others if it were possible to use a clonal selection method, but we expect this effect to be relatively minor and also believe that producing reproducible data with multiple heterogeneous populations of transduced cells minimises the chance of observing off target effects from one specific clone. It is more likely that a higher MOI would have yielded cells capable of expressing higher levels of induced wild type *NDUFC2*, however this also increases the number of integration events and therefore increases the chances of off target effects by disrupting other genes and increases the chances of the expression being too high, which we have found problematic in our experience with other genes we have investigated. We therefore decided to stay with our approach as this demonstrates that even a relatively modest increase in wild type

NDUFC2 expression in the fibroblast cell lines from affected individuals can clearly improve the CI defect, despite not reaching control levels, which provides strong evidence for the pathogenicity of these *NDUFC2* variants.

Referee #2 (Remarks for Author):

Authors describe two families with biallelic variants in *NDUFC2* gene leading to an isolated complex I enzyme deficiency and a confirmed defect in the assembly of the complex I holoenzyme. One individual from family 1 harboring a homozygous missense variant involving a highly conserved residue and second individual from family 2 a homozygous loss-of-function variant. Authors have performed sufficient amount of functional studies to prove the pathogenicity of these variants in *NDUFC2* gene. There is one weak point, authors have not investigated individual S2 in Family 1. But I understand that it is always not possible to get the material from deceased individuals.

Author's Response: We thank the reviewer for their helpful comments on our manuscript. We are able to confirm, as stated above in response to the previous reviewer's comments, that the reason Subject 2 from Family 1 was not investigated in detail was due to a lack of available biological samples, namely a primary fibroblast cell line. We have tried to make this clearer throughout the manuscript (specific points listed in response to previous reviewer's comments above) and acknowledge that while we do describe three subjects clinically, only two of these were subject to a full, functional characterisation.

Minor corrections:

I suggest to use word "variant" or pathogenic variant" instead of mutation in the title. See section Terminology:https://www.acmg.net/docs/Standards_Guidelines_for_the_Interpretation_of_Sequence_Variants.pdf

Author's Response: We thank the reviewer for pointing this out; we agree and have now changed this.

Introduction:

Please write out abbreviation CSF

Author's Response: This has now been added.

Page 6, Case reports:

I suggest to write out "Supplementary figure 1a and 1b, etc or Supp figure" throw-out the manuscript. The abbreviation "Figure EV" is confusing. Or please give an explanation for EV.

Author's Response: This is the journal's designated nomenclature, the 'EV' stands for 'expanded view' and is the equivalent of supplementary data. For this reason, we are not able to modify the title of these figures.

Please add to Family 1 and 2 the citation to Figure 1a - pedigrees of both families.

Author's Response: Thank you for pointing this out, this has now been done.

Page 8, Molecular genetic investigations:

Para: "Quantitative real-time PCR (qRT-PCR) analysis of fibroblast-derived mRNA transcript levels demonstrated decreased expression of *NDUFC2* mRNA in Subject 1 fibroblasts (43% of controls), while mRNA levels in Subject 3 fibroblasts were comparable to controls (Figure 2A)."

I do not understand how Figure 2A express these results? Please correct "Figure 2D".

Page 9, Subject fibroblasts and skeletal muscle display isolated complex I deficiency:
Here is the same. The citation of Figure 2 subunits is not right. Please correct.

Author's Response: We would like to thank the reviewer for highlighting the incorrect referencing within Figure 2. As we noted in response to the previous reviewer's comments, we have now rearranged the Figure panels so that Fig 2A shows the qRT-PCR data. We apologise for this oversight on our part.

4th Aug 2020

Dear Prof. Taylor,

Thank you for the submission of your revised manuscript to EMBO Molecular Medicine. I am pleased to inform you that we will be able to accept your manuscript pending the following final amendments:

1) In the main manuscript file, please do the following:

- In addition to the accession number please provide URL for your deposited data. We noted that the "PXD014936" is not freely accessible. Please be aware that all datasets should be made freely available upon acceptance, without restriction. That applies also to NGS data generated in this study. Use the following format to report the accession number of your data:

**EMM-2020-12619V3, Alahmad *et al.*
Response to editor's comments.**

Thank you for the submission of your revised manuscript to EMBO Molecular Medicine. I am pleased to inform you that we will be able to accept your manuscript pending the following final amendments:

Response: Thank you, we will note our response to each individual point below.

1) In the main manuscript file, please do the following:

- In addition to the accession number please provide URL for your deposited data. We noted that the "PXD014936" is not freely accessible. Please be aware that all datasets should be made freely available upon acceptance, without restriction. That applies also to NGS data generated in this study. Use the following format to report the accession number of your data:

The datasets produced in this study are available in the following databases:
[data type]: [full name of the resource] [accession number/identifier] ([doi or URL
or identifiers.org/DATABASE:ACCESSION])

Response: The deposited data was not accessible until acceptance; this has now been changed and the information is now freely available on the website that we cite; the URL for access to the complexome data has now been included in the manuscript in the designated format. Unfortunately the patient consent obtained does not allow us to make the raw WES data available online as noted in the author checklist document.

The authors performed the requested changes.

Corresponding Author Name: Robert W. Taylor
Journal Submitted to: EMBO Molecular Medicine
Manuscript Number: EMM-2020-12619